# OpenReview forum: "On the Safety of Open-Sourced Large Language Models: Does Alignment Really Prevent Them From Being Misused?"
_ICLR.cc/2024/Conference — ICLR 2024 Conference Withdrawn Submission_

### Official Review · Reviewer_ib1R · 2023-10-12

**Soundness:** 1 poor
**Presentation:** 2 fair
**Contribution:** 1 poor
**Rating:** 3
**Confidence:** 4

**Summary:**

This paper addresses the question of whether is possible to manipulate open-sourced LLMs to achieve the generation of undesired content. The authors propose a method called Probability Manipulation (ProMan) that intervenes in the generation process of LLMs to guide them in generating harmful or sensitive content. The key idea behind ProMan is to directly manipulate the probability distribution of tokens in the generation process of the LMs to guide them in generating specific tokens, in order to create an affirmative prefix. Secondly, they alternate specific tokens during the generation with their predefined negative pairs (e.g. unethical → ethical). They conduct experiments on exposing harmful content and on private leakage with four open-sourced LLMs and demonstrate various case studies of undesired generations. The paper also discusses potential countermeasures to mitigate harmful generations. The contributions of this work include the introduction of ProMan, the empirical evaluation of its performance, and the exploration of potential defence strategies for open-sourced LLMs.

**Strengths:**

1) The concept of manipulating the probability distribution in order to produce affirmative prefixes and negation reversing, while unesecery in the specific case, is indeed interesting. Negation reversing and token alternations through manipulating the probability distribution, in general, should be further explored.

2) The quality of the paper is generally good, with a well-structured presentation of the problem, methodology, and experimental results. The experiments conducted on different LLMs and attack scenarios provide initial evidence for the effectiveness of ProMan in generating undesired content and privacy leakage.

3) By demonstrating the ability to manipulate LLMs and generate harmful content, the paper raises important questions about the security and responsible use of these models. It contributes to the ongoing discussions on LLM safety and calls for more advanced mitigation strategies.

**Weaknesses:**

1) The authors promise to misguide open LLMs "without careful prompt designs". They do exactly that, in a careful and complicated manner they design the prompt of the model and alternate specific tokens of the produced output (that is used as input to produce the following tokens...).

2) The interesting probability manipulation idea, in practice consists of enforcing an affirmative prefix and negating six predefined tokens.

3) The enforced production of the affirmative prefix,  is identical to appending a suffix to the prompt, making it unnecessarily complicated and less novel than what is presented.

4) The negation reversing aspect, while the only novelty of the paper, is limited in scope as it only changes six tokens with their predefined negative pairs.

5) The paper only compares ProMan with two baseline attacks, one heuristic and one optimization-based, but fails to provide a comprehensive evaluation against a wider range of existing attack methods.

6) It would be beneficial to expand the evaluation to larger LLMs to provide more robust and reliable results on scale.

7) For the privacy leakage experiment the evaluation is based on a relatively small evaluation set of 100 queries.

8) In the Possible Countermeasures section the paper presents almost generic methods from the literature to address current issues with LLMs. It would be beneficial to discuss specific solutions for the specific proposed method or for "prompt hacking" in general.

9) The paper lacks a thorough analysis and discussion of the potential ethical implications of the proposed method. Given that the goal of the work is to expose the limitations of alignment in preventing misuse of LLMs, it would be important to address the potential harm and ethical concerns associated with such misuse. Providing a more comprehensive analysis of the ethical considerations and discussing potential mitigation strategies would strengthen the paper's contribution to addressing the responsible development and usage of LLMs.

**Questions:**

1. How scalable is ProMan? Have the authors considered evaluating the method on larger LLMs to determine its effectiveness and robustness on a larger scale?

2.  In the Possible Countermeasures section, the authors present generic methods from the literature. Can they discuss specific solutions or potential countermeasures that are more tailored to address the proposed method of ProMan or the broader concept of "prompt hacking"?

3.  Given that appending an affirmative prefix to the prompt achieves a similar outcome as ProMan, can the authors explain the specific advantages or novelty of ProMan in generating the affirmative prefix? How does ProMan go beyond what can be achieved through simpler methods?

4. Can the authors provide insights into the potential impact of ProMan on the alignment of LLMs and the implications for the development and usage of open-source LLMs? How can the findings of this work contribute to improving the alignment and safety measures of LLMs in the future?

5. Are there any limitations or challenges that the authors encountered during the implementation or evaluation of ProMan that may impact its practicality or effectiveness?

**Details Of Ethics Concerns:**

One concern regarding the ethical considerations in this paper is the potential harm that can arise from the misuse of LLMs. The paper demonstrates how ProMan can be used to generate harmful, biased, or privacy-sensitive content. While the authors have taken precautions to obscure potentially biased content in the examples, there is still the risk that these techniques could be used to generate and spread harmful or misleading information.

Additionally, the paper does not provide an in-depth analysis or discussion of the ethical implications associated with the proposed method.

---

### Official Review · Reviewer_8iMJ · 2023-10-29

**Soundness:** 2 fair
**Presentation:** 2 fair
**Contribution:** 1 poor
**Rating:** 3
**Confidence:** 3

**Summary:**

The paper investigates the question how to best attack an LLM so that it generates either harmful responses or leak personal information. To achieve this, the paper proposes to leverage the affirmative prompt as well as negation reversion to achieve this.

Empirically, the authors show that their method achieve best attach performance for generating harmful responses as well as leaking personal information compared to both heuristic baselines and optimization baselines. The ablations study show that affirmative prompt and negation reversion are both important to achieve this result.

**Strengths:**

The authors propose an attack that combine affirmative prompting (that people have proposed before) and negation reversion (which seems to be the novelty of this paper). This combination have shown good results in two datasets and the authors have compared with various other approaches for this important question.

**Weaknesses:**

The novel negation reversion approach is implemented based on "negative words" used which limits its applicability and wide adoptions.

In introduction and methodology section, the paper advocates the "Proman" approach which alters the output token probability to enable the attack. However, after reading the paper, neither affirmative (which can be realized through prompting) and negative reversion (which can be realized through token substitution) really needs the probability manipulation.

**Questions:**

The authors said "..affirmative prefix which only impact the first few tokens, negation reversing potentially impacts every token in the response". Can authors provide citations or experimental results for this claim please? This doesn't seem an intuitive claim, at least for me.

---

### Official Review · Reviewer_cQgR · 2023-10-31

**Soundness:** 2 fair
**Presentation:** 3 good
**Contribution:** 2 fair
**Rating:** 5
**Confidence:** 4

**Summary:**

- This paper proposed ProMan, a method to white-box attack LLMs to generate harmful responses.
- They focus on white-box attacking an "aligned" LLM and trying to "de-align" the model.
- They proposed PM and AP, where PM means manipulating the output distributions of the LLMs and AP stands for adding affirmative prefix to make the LLM “agree” to generate expected harmful content.
- Experiments showed that ProMan is better than heuristic attacks and optimization-based attacks.

**Strengths:**

- ProMan achieved higher attack success rates than heuristic and optimization-based attacks.
- Simple but effective attack method requiring no careful prompt engineering or heavy computation.

**Weaknesses:**

- The setting this paper presents is a little bit weird to me. Normally, the developers will not deploy applications that allow the users to modify the output distribution of it when decoding. The only possible use case for this paper is when a developer downloads the open-sourced model and intends to extract some of the harmful content from the LLM. I don't think the developer would need this method at all, as they can simply choose the "unaligned" LLMs (e.g. LLaMA v1 without alignment) to generate the harmful content. If you want to generate something harmful, what's the point of getting yourself into trouble by using an aligned version of LLM and trying hard to circumvent/de-align the alignment?
- A more desirable setting will be attacking the aligned LLMs using only input prompts without the need to modify the output distributions. It would be a broad scenario that works for all the deployed real-world applications for public users. But this paper is not in this direction.

**Questions:**

- Is it possible to extend this ProMan method to the case of black-box setting? It seems that the affirmative prefix trick itself -- or "ProMan w/o NR" -- can be applied without white-box access.  But the result shows that "ProMan w/o NR" is not that good in terms of performance. Can you discuss more about the necessity of white-box access in your method?

---

### Official Review · Reviewer_ssVq · 2023-11-01

**Soundness:** 2 fair
**Presentation:** 3 good
**Contribution:** 2 fair
**Rating:** 3
**Confidence:** 4

**Summary:**

The paper presents a simple methods that allows for generating harmful content using open-source LLMs. The method assumes access to the decoding internals of the model, e.g., it changes the logits computed for selected output vocab. The method consists of two components: 1) forcing the model to start with an affirmative prefix (e.g., "Sure, here is") and it does so by adding a large enough positive constant to the logits corresponding to the tokens of the prefix, 2) Reversing negative words and it does so by maintaining a list of negative words (e.g., Illegal) that are detected during the decoding process and replaced with the antonyms. Evaluation on the AdvBench benchmark shows an advantage of the presented method over black-box attacks in terms of both the affirmativeness and harmfulness of the generated outputs.

**Strengths:**

1. The paper shows that open-source LLMs can very easily be used for generating harmful content via white-box attacks.

**Weaknesses:**

1. It is unclear why "Probability manipulation" is needed for the presented approach. Simply force-decoding an affirmative prefix and force-replacing negative words would be an equivalent approach. The paper needs to justify the introduction of the probability manipulation approach (equations 1 & 2).

2. Evaluation of affirmativeness and harmfulness is conducted automatically via word matching and an external LLM, respectively. While LLM-based evaluation can be accurate, still some human evaluation is critically needed for this work. For example, affirmativeness is judged based on the presence/absence of a fixed set of negative words. Is that set of words comprehensive enough? How about False positives? Similarly, the quality of the used LLM for judging harmfulness needs to be confirmed even with a small set of human verified examples.

3. The paper lacks error analysis. Table 1 for examples shows that the presented approach is still not perfect. The paper needs to look into the failure modes.

**Questions:**

NA